# Experimental Study of Mechanical Properties and Failure Characteristics of Coal–Rock-like Composite Based on 3D Printing Technology

**DOI:** 10.3390/ma16103681

**Published:** 2023-05-11

**Authors:** Ying Chen, Zikai Zhang, Chen Cao, Shuai Wang, Guangyuan Xu, Yang Chen, Jinliang Liu

**Affiliations:** 1College of Mining, Liaoning Technical University, Fuxin 123008, China; 2Research Center for Rock Burst Control, Shandong Energy Group Co., Ltd., Jinan 250014, China; 3Huafeng Coal Mine, Xinwen Mining Group Co., Ltd., Tai’an 271413, China

**Keywords:** 3D printing, coal rock, composite, failure mode, impact propensity

## Abstract

Coal contains cracks and has strong heterogeneity, so the data dispersion is large in laboratory tests. In this study, 3D printing technology is used to simulate hard rock and coal, and the rock mechanics test method is used to carry out the coal–rock combination experiment. The deformation characteristics and failure modes of the combination are analyzed and compared with the relevant parameters of the single body. The results show that the uniaxial compressive strength of the composite sample is inversely proportional to the thickness of the weak body and directly proportional to the thickness of the strong body. The Protodyakonov model or ASTM model can be used as a verification method for the results of a uniaxial compressive strength test of coal–rock combination. The elastic modulus of the combination is the equivalent elastic modulus, and the elastic modulus of the combination is between the elastic modulus of the two constituent monomers, which can be analyzed using the Reuss model. The failure of the composite sample occurs in the low-strength material, while the high-strength section is rebounding as an extra load on the low-strength body, which may cause a sharp increase in the strain rate of the weak body. The main failure mode of the sample with a small height–diameter ratio is splitting, and the failure mode of the sample with a large height–diameter ratio is shear fracturing. When the height–diameter ratio is not greater than 1, it shows pure splitting, and when the height–diameter ratio is 1~2, it shows a mixed mode of splitting and shear fracture. The shape has a significant effect on the uniaxial compressive strength of the composite specimen. For the impact propensity, it can be determined that the uniaxial compressive strength of the combination is higher than that of the single body, and the dynamic failure time is lower than that of the single body. It can hardly determine the elastic energy and the impact energy of the composite with the relationship to the weak body. The proposed methodology provides new cutting-edge test technologies in the study of coal and coal-like materials, with an exploration of their mechanical properties under compression.

## 1. Introduction

A coal bump is a kind of geo-dynamic disaster that occurs in underground coal mines. It is sudden and destructive, causing casualties and economic losses [1,2,3]. It is understood that a coal bump is involved with the roof and floor of a coal seam. Therefore, research on deformational behavior and impact characteristics of the roof–coal–floor combination has been carried out since the 1990s [4]. The mechanical properties of coal–rock composite specimens were studied based on an experimental method, and the failure mode and impact performance of the composite sample were evaluated under different coal and rock strengths and coal–rock proportions [1,2,3,4,5,6,7,8,9,10,11,12,13,14,15,16].

The early research mainly predicted the impact hazard of the composite sample through the mechanical properties of the coal and rock composite, including the average elastic modulus, Poisson’s ratio and compressive strength. The results show that the impact index predicted by the composite is higher than that of the coal sample, and the larger the rock proportion is, the stronger the impact propensity is [1,2,3,4,5,6,7,8,9,10,11,12]. It is also found that the impact tendency of the composite increases with the increase in the strength of the roof and floor [10,11,12,13,14,15,16].

The failure evolution and energy accumulation and dissipation processes of a coal–rock composite under compressive pressure are research hotspots [11,12,13,14,15,16]. The deformational behavior of a coal–rock composite is dominated by the randomly distributed cracks generated from coal seam formation. The cracks generated within the coal body connect with each other, forming macro-fracture, extending to the rock section with the increasing load [15,16]. The process of fracture development can be divided into four stages: pore compaction, crack generation and development, fracture development and penetration, and fracture failure [16,17,18,19]. The failure process of coal and rock composites is also verified through the use of computer simulation results [19].

From the perspective of energy accumulation and dissipation, the bearing process of the coal–rock composite can be divided into three stages: rapid accumulation of strain energy, slow growth of strain energy, and rapid release of strain energy [18,19]. When the strength and proportion of the rock section increase, the strain energy density of the composite increases rapidly and then decreases slowly [19]. As a result, the failure duration of the composite is reduced, and the dynamic behavior during failure is enhanced. From the viewpoint of practice, the greater the roof strength and thickness, the higher the coal seam impact tendency. Some scholars believe that the energy change rate while coal–rock composite failure is a more accurate indicator of the coal impact propensity and should be included in the conventional rockburst prediction method [20].

In 2020, Du [21] reviewed and summarized the results of important research on coal–rock composite studies for 3 decades. This review article discusses the deformational characteristics, burst proneness and disaster prevention of coal–rock composite from the aspects of sampling, combination mode, and impact energy index. It is also pointed out that the variation in experimental data of coal–rock composite research is large, which introduces difficulties into the engineering practice of burst risk assessment.

Coal and rock are typical heterogeneity and anisotropic materials and are often inelastic. The geological conditions of coal mines are varied, and the control factors of coal seam formation are quite different. It leads to poor consistency in the experimental data when using raw coal samples. It shows that the experimental results may be quite different even if all samples were drilled from one coal block. In addition, for coal–rock composite research, field sampling is often time-consuming and laborious, as three kinds of materials are used for a composite sample. As a result, there often only a few samples were tested in a coal–rock composite experimental study.

Structural or coal-like materials are often used to simulate raw coal in a laboratory. The most commonly used materials to model coal is often selected from gypsum, cement and coal powder. It is significant to develop new coal-like materials with an exploration of their structural characteristics and mechanical properties. In recent years, the rapid development of 3DP materials has allowed for a wide range of mechanical properties, and the experimental results are highly repeatable. In this study, 3DP material is used to study the deformation and failure law of coal–rock composite for the first time. Vero White Plus photosensitive resin is used to simulate the hard rock in the roof and floor, and the coated sand is used to simulate the coal body. The selected 3DP materials have the mechanical properties of the coal and rock in typical burst coal mine. By analyzing its burst propensity index and comparing it with a single material sample, the deformational characteristics and impact propensity are obtained, which provides a reference for coal–rock composite research. The 3DP material overcomes the shortcomings of the small number of samples and the low repeatability of experimental results in similar studies. The proposed methodology provides a new cutting-edge test technology in the study of coal and coal-like materials, with an exploration of their mechanical properties under compression.

## 2. Materials and Methods

### 2.1. Sample Preparation

In coal–rock composite research, the sample includes the roof, floor, and coal, and it is necessary to select 3DP materials that can simulate hard rock and a soft coal seam. Based on the mechanical parameter test of 3D printing materials [22], Vero White Plus photosensitive resin, made by the Material laboratory at Liaoning Technical University, China, was selected to simulate hard rocks, and coated sand was selected to simulate a soft coal seam. Vero White Plus material samples are prepared through the Stereo lithography Appearance (SLA) method, with a layer thickness of 0.1 mm and an accuracy of 0.02 mm. The coated sand sample is prepared through the use of a laser sintering printer, and the maximum particle size is 0.21 mm.

The samples were prepared and tested according to the International Society of Rock Mechanics (ISRM) recommended testing method [23,24]. The diameter of the prepared sample is 50 mm, and the total height is 100 mm. The Vero White Plus single sample height is 10 mm, 17.5 mm, 33 mm, 40 mm, 50 mm, and 100 mm; The height of coated sand single sample includes 20 mm, 33 mm, 50 mm, 66 mm, 80 mm, and 100 mm. Vero White Plus: coated sand: Vero White Plus is 1:1:1, 2:1:2, 1:4:1, and 1:8:1 in the three structural samples formed after the combination of the above single samples, and Vero White Plus: coated sand is 1:1 in the two structural samples. The interface of the composite sample is in natural contact. The prepared monomer sample and the combined sample are shown in Figure 1 and Figure 2. The height difference between the end face of the sample is 0.02 mm.

### 2.2. Mechanical Properties of Materials

The WAW-600C microcomputer-controlled electro-hydraulic servo universal testing machine made by Jinli Laboratory Instruments, Changchun, China, is used for the uniaxial compression test using the ISRM suggested testing method. The loading mode is displacement loading, and the loading rate is 1 mm/min. The strain gauge is adhered to the middle of the sample to measure the radial and axial strains of the sample. The uniaxial compression stress–strain curve of the Vero White Plus specimen is shown in Figure 3. The material has a clear linear elastic stage under uniaxial compression and enters the softening stage after reaching a peak strength of about 120 MPa. The mechanical parameters of the material have a good consistency.

The uniaxial compression stress–strain curve of the standard sample of coated sand is shown in Figure 4. The stress–strain curve is similar to that of a raw coal sample. Compaction, elastic deformation, and fracture stages occur before the peak strength. After reaching a peak of about 10 MPa, there is an evident drop in stress, showing post-peak strain characteristics with certain brittle failure characteristics.

The test results of the standard Vero White Plus and coated sand samples are shown in Table 1. In the table, UCS is the uniaxial compressive strength, *E* is the elastic modulus, and ν is Poisson’s ratio.

Table 1 shows that the data of Vero White Plus and the coated sand samples have high consistency, which conforms to the expectations of the application of 3DP materials to rock mechanics engineering. According to the magnitude of UCS and *E*, Vero White Plus can better simulate hard sandstone, and the mechanical parameters of the coated sand are close to that of hard raw coal. It should be noted that, as Vero White Plus is a polymer base material, its Poisson’s ratio is higher than that of sandstone. In some experiments or numerical calculations that have a large influence in terms of Poisson’s ratio, appropriate adjustments should be made according to the problems studied, such as experiments involving confining pressure and simultaneous tension and compression, or calculations using bulk modulus, shear modulus, and Lame constant.

## 3. Results and Discussion

### 3.1. Mechanical Characteristics of Coated Sand with Different Height–Diameter Ratio

The height–diameter ratio is a benchmark for the study of a composite material as it affects the strength and failure mode of the material. To study the effect of the height–diameter ratio on 3DP-coated sand and compare it with typical hard coal, the uniaxial compression test of six coated sand monomers with different height–diameter ratios was carried out. The stress–strain curves of various height–diameter ratio specimens are shown in Figure 5, and the results are summarized in Table 2.

The experimental results show that the uniaxial compressive strength, elastic modulus, and peak axial strain of the coated sand samples with different height–diameter ratios are different. The height–diameter ratio of the sample has a significant influence on its uniaxial compressive strength. As the height–diameter ratio of the sample increases from 0.4 to 2.0, the uniaxial compressive strength decreases from 15.18 MPa to 10.02 MPa, showing a linear decrease (Figure 6), and the correlation coefficient is 0.90. The uniaxial compressive strength at the aspect ratio of 2.0 is 27% lower than that at the aspect ratio of 1.0, which is close to the research results of Townsend et al. [25], that is, the uniaxial compressive strength of the sample with the aspect ratio of 2 is about 20%~30% lower than that of the sample with the aspect ratio of 1. As the height–diameter ratio increases from 0.4 to 2.0, the elastic modulus of the sample increases from 0.353 GPa to 1.189 GPa. The sample with a height–diameter ratio of 1 is 77% of the sample with a height–diameter ratio of 2 (Figure 6b). It is similar to the size effect of natural rocks [26,27].

### 3.2. Failure Characteristics of Coated Sand with Different Height–Diameter Ratio

The failure characteristics of 3DP-coated sand monomer samples with different height–diameter ratios after the uniaxial compression test are shown in Figure 7. It can be seen that the primary failure modes of the sample are splitting and shear fracture, which agrees well with the failure mode using a raw coal sample. The splitting failure of the samples with height–diameter ratios of 0.4, 0.66, and 1 all run through the upper and lower ends of the sample. The samples with height–diameter ratios of 1.3 and 1.6 show a mixture of splitting and shear fracture. The samples with a height–diameter ratio of 2 show a typical shear fracture pattern. In general, with the increase in height–diameter ratio, the failure mode of the sample changes from splitting failure through the upper and lower ends to shear failure, and the samples with height–diameter ratios of 1.3 and 1.6 belong to a mixed fracture mode.

### 3.3. Uniaxial Compressive Strength and Elastic Modulus of Composite Sample

The uniaxial compressive tests were conducted using a composite sample of Vero White Plus and coated sand. The experimental results are shown in Table 3. In the table, VWP represents Vero White Plus, CS represents coated sand, the order of materials represents the composition of materials from top to bottom in the composite sample, and the proportion in brackets represents the thickness proportions of different materials. The stress–strain curve of the composite specimen under uniaxial compression is shown in Figure 8.

Figure 9 shows the relationship between the uniaxial compressive strength, elastic modulus, and the thickness of the coated sand in the composite. It can be seen from Figure 9a that the uniaxial compressive strength of the composite sample is inversely proportional to the thickness of the coated sand and is directly proportional to the thickness of Vero White Plus. As the proportion of coated sand in the composite sample increases, the uniaxial compressive strength of the composite sample gradually decreases. The uniaxial compressive strength of the sample with a thickness of 20 mm of the coated sand is 18.25 MPa, and the uniaxial compressive strength of the sample with a thickness of 80 mm of coated sand is 9.50 MPa, showing a similar change to that of the natural coal–rock combination; that is, when the thickness of the monomer with lower strength in the combination increases, the uniaxial compressive strength of the coal–rock combination decreases [2,17,18,19]. It can also be seen from the figure that under the same thickness of coated sand, the uniaxial compressive strength of the combination is higher than that of the monomer, especially when the thickness of coated sand is small.

It can be seen from Figure 9b that the elastic modulus of the combination is inversely proportional to the thickness of the coated sand. With the increase in the proportion of the coated sand in the combination, the elastic modulus of the combination gradually decreases. When the proportion of coated sand in the combination reaches 80%, the elastic modulus of the combination is basically the same as that of the coated sand monomer with a height of 100 mm.

### 3.4. Failure Characteristics of Composite Sample

The typical failure modes of the composite sample are shown in Figure 10, which are similar to previous composite tests using geo-materials. It can be seen that the fracture surface of the coated sand samples with a thickness of 20 mm~50 mm in the combination is mainly split through the upper and lower ends, which is consistent with the fracture characteristics of the single coated sand sample with a height–diameter ratio of 0.4 (20 mm high), 0.66 (33 mm high) and 1 (50 mm high). The specimen with the thickness of 65 mm and 80 mm of coated sand in the combination formed the shear failure of the non-penetrating end surface, which was consistent with the fracture characteristics of the single specimen of coated sand with the height–diameter ratio of 1.3 (height 66 mm) and 1.6 (height 80 mm). The experiment shows that the fracture characteristics of the composite sample with the same thickness of coated sand are basically consistent with that of the single sample. At the same time, it is also shown that under the condition of the uniaxial compression test, the relatively weak monomer shape of coated sand is the key to controlling the failure characteristics of the composite. Since the strength of Vero White Plus is much higher than that of coated sand, there is no fracture in this test.

### 3.5. Deformation and Failure Characteristics of Composite Sample

#### 3.5.1. Stress–Strain of the Combination

The deformation characteristics of the specimen are determined by its stress distribution. Thuro et al. [28] proposed a conceptual explanation of the shape effect, pointing out that the stress and strain in different regions of the specimen with a height–diameter ratio of 1:1 and the specimen with a height–diameter ratio of 3:1 are significantly different, which determines the mechanical response of the specimens. This interpretation still applies to the combination. Figure 11 shows the comparison of the stress and strain of composite and single samples with the same height–diameter ratio.

Ver White Plus and coated sand in Figure 11a have a height–diameter ratio of 1:1 and form relatively independent stress and strain distribution under uniaxial compression. The failure pattern of coated sand is consistent with that of coated sand with a height–diameter ratio of 1. The height–diameter ratio of the coated sand sample in Figure 11b is 2, and its stress and strain state is completely different from the composite sample in Figure 11a. This difference determines the deformation strength characteristics and failure mode of the combination.

#### 3.5.2. Uniaxial Compressive Strength of Composite

Almost all the studies of the combination think that the uniaxial compressive strength of the combination is higher than that of the coal sample and lower than that of the rock [21]. The higher the height of the rock in the sample, the greater the strength of the combination. The results of this study also support this conclusion. The decisive factor of the strength change of the combination is the height of low-strength coated sand. With the decrease in the proportion of coated sand, the strength of the combination will increase.

In theory, the combination is a series of multiple materials, and the axial stress of different parts is the same under uniaxial compression test conditions. Therefore, in the process of uniaxial compression, the compressive strength of the combination is, in fact, the uniaxial compressive strength of the medium with the lowest strength among all media. The uniaxial compression test of the coated sand monomer in Section 3 and the uniaxial compression test of the composite in Section 4 show that the strength of the composite at the same height is higher than that of the coated sand monomer at the same height. The main influencing factors are the shape effect and end effect [29,30,31]. The shape has a significant impact on the uniaxial compressive strength of the composite sample, and the effect of the end effect is relatively small. Therefore, in the study of the composite, if the uniaxial compressive strength of the monomer with the corresponding specifications in the composite can be measured, then the uniaxial compressive strength of the composite can be predicted.

Protodyakonov [32] established the relationship between the uniaxial compressive strength of standard and non-standard specimens with a height–diameter ratio of 2, as in the following equation:(1)$${\mathrm{UCS}}_{2}=8\mathrm{UCS}/(7+2D/L)$$
where *UCS*_2_ is the uniaxial compressive strength of the standard sample converted to the height–diameter ratio of 2, and *UCS* is the uniaxial compressive strength of the non-standard shape sample measured, *D* is the diameter of the test specimen and *L* is the height of the test specimen. When the height–diameter ratio of the test sample is 1 to 3, the variation range of UCS is 0.89 to 1.04 [28].

ASTM [31] proposes that the conversion relationship between the uniaxial compressive strength of non-standard specimens with a height–diameter ratio of less than 2 and standard specimens with a height–diameter ratio of 2 is as seen in the following equation:(2)$${\mathrm{UCS}}_{2}=\mathrm{UCS}/(0.88+0.24D/L)$$

According to the empirical formula of Equations (1) and (2), the calculation results and laboratory measurement results are shown in Figure 12. The difference between the calculated results of the empirical formula and the laboratory measurement results is 0.88~1.21. The results of the literature [2,17,19] are also in good agreement with those calculated by the above relationship.

The strength of the combination is higher than that of the monomer, and the strength of the combination can be estimated by the Protodyakonov model or ASTM model [33]. This can be used as a verification method of uniaxial compressive strength test results in the study of coal–rock combination. In addition, because the coal contains cracks of different scales and has strong heterogeneity when measuring its uniaxial compressive strength in the laboratory, the data obtained are often more discrete, which should be considered in the research.

#### 3.5.3. Elastic Modulus of Composite

The uniaxial compression test of the combination is a stress problem under the condition of a binary medium in series, and its elastic modulus should be the equivalent elastic modulus on the Reuss hypothesis, as expressed in Equation (3) [34]. Reuss assumes that the strain fields of various minerals that make up the rock are not uniform, but they all bear the same stress. The effective elastic modulus of composite based on the Reuss hypothesis is as follows:(3)$${E=(f}_{1}{/E}_{1}{+f}_{2}{/E}_{2}{)}^{-1}$$
where *f_1_* and *f_2_* are the volume integrals of each monomer and *E_1_* and *E_2_* are the elastic moduli of the monomer.

According to Equation (3), in the elastic deformation stage of the combination, the elastic modulus of the combination is related to the elastic modulus of each monomer and its proportion. Figure 13 shows the comparison between the experimental measurement results of the elastic modulus of different combinations in this study and the theoretical calculation results. It can be seen from the figure that the theoretical results and experimental results are in good agreement.

The elastic modulus of the composite is between that of the two constituent monomers. In the sample composed of coal with a low elastic modulus and a rock with a high elastic modulus, it can be determined that the elastic modulus of the combination will be higher than the elastic modulus of coal, lower than the elastic modulus of rock, and it will increase with the increase in the rock volume/height ratio. Table 4 shows the comparison data of Formula (3) in [17,19], and the results show that the calculated value is very close to the measured value. The elastic modulus of the composite can be estimated according to the Reuss model. The heterogeneity of middling coal in the experiment has a great influence on the consistency of the results.

#### 3.5.4. Failure Characteristics of Combination

In general, the failure of the composite specimen mainly occurs in the low-strength part, and the failure mode of this part is also related to its shape [21]. According to the experimental results, the failure modes of the sample can be divided into two modes. The main failure mode of the sample with a small height–diameter ratio is splitting, and the failure mode of the sample with a large height–diameter ratio is a shear fracture. When the height–diameter ratio is not greater than 1, the fracture often appears as pure splitting. When the height–diameter ratio is 1~2, the fracture mode is a mixed mode of splitting and shear fracture. The greater the height–diameter ratio, the more shear fracture characteristics.

When the strength of the two parts of the combination is not different, because of the end effect, in the process of the weak body failure, the local part of the strong body may also reach the fracture condition and cause failure. This kind of phenomenon occurs in the study of coal and rock mass combinations with low rock strength. The reason is that the stronger part is destroyed under high stress at the end face.

The monomer has different mechanical properties, and the interaction between them in the loading process affects the failure characteristics of the composite. In this study or similar studies, the low-strength part fails first, and the high-strength parts undergo elastic rebound correspondingly. The elastic rebound also acts as a load on the fracture body, which may cause a sharp increase in the strain rate of the fracture body, resulting in the characteristics of dynamic failure. In this study, the failure process after setting a spring with a rigidity of 20 KN/mm between the sample and the indenter of the testing machine is shown in Figure 14. The time shown in the figure is relative time. It can be seen that the failure process of the coal sample is only 0.52 s, and the whole coal sample is washed out after failure.

In the combination test in this study, the rigidity of the testing machine is not less than 5 MN/mm, and the rigidity of Vero White Plus is 53 KN/mm, which is nearly 100 times different. The rigidity of the coal and rock measured in the laboratory is about n × 10^2^ kN/mm~n × 10^3^ kN/mm. From this experiment and the uniaxial compression failure tests of other combinations, there is almost no dynamic failure.

#### 3.5.5. Energy Characteristics of Combination

As for the comparison between the elastic energy index and the impact energy index of the composite and the monomer, the results are different in the literature. Li et al. [3] show that both the elastic energy index and impact energy index increase or decrease. Dou et al. [4] reported that the impact energy index of the combined coal and rock samples decreased with the increase of the percentage of coal samples, and the elastic energy index gradually increased with the increase in the percentage of coal samples. Lu et al. [5] believe that the elastic energy index and impact energy index of the combination are higher than that of the monomer. Zuo et al. [7] believe that as the strength of the combination increases, the pre-peak accumulated energy and post-peak dissipated energy increase and the impact energy index increases. Li et al. [15] used structural rock material to show that the impact tendency of the combination is lower than that of single coal and single rock.

Figure 15 shows the energy comparison between the combination and the monomer under typical uniaxial compression. It can be seen that the relationship between the energy before and after the peak strength of the standard coated sand monomer and the combination (coated sand:Vero White Plus is 1:1) cannot be determined.

### 3.6. Burst Proneness of Composite

According to the determination method of the coal impact propensity index, the comparison results of combination and monomer are calculated as shown in Table 5.

According to the experimental results, the uniaxial compressive strength of the coal–rock combination is higher than that of coal. After reaching the peak strength, the strain of the middling coal body in the coal–rock combination increases, and the stress decreases instantaneously, and the rock rebounds elastically, which provides the load to the coal body instantaneously and accelerates the destruction of the coal. This is equivalent to the formation of a non-rigid press, and the dynamic destruction time of the combined coal–rock body is obviously smaller than that of the single body.

The elastic energy index is determined by the stress and strain of the sample. Since the elastic modulus of the coal–rock composite sample is higher than that of the coal body, the strain is lower than that of the coal body, and the relationship between the elastic energy index of the coal–rock composite and the coal body cannot be determined. The impact energy index is the ratio of the deformation energy accumulated before the peak value to the deformation energy lost after the peak value in the whole stress–strain curve of the specimen under uniaxial compression. This value increases the energy consumption of failure on the basis of the elastic deformation energy, which is more complex and cannot be easily identified.

## 4. Conclusions

In this study, 3D printing technology is used to prepare coal-and-rock-like samples. Based on uniaxial compressive testing, the deformation characteristics and failure modes of the composite and their relationship with the individual are analyzed. The impact propensity index of the composite and that of the individual is further discussed. The proposed methodology provides new cutting-edge test technologies in the study of coal and coal-like materials, with an exploration of their mechanical properties under compression. The main conclusions are as follows:(i)The uniaxial compressive strength of the composite sample is inversely proportional to the thickness of the weak body and directly proportional to the thickness of the strong body. It is mainly controlled by the shape of the weak body. The results of the uniaxial compressive strength of the composite body can be verified and predicted using the Protodyakonov model or ASTM model. The elastic modulus of the composite is between the elastic moduli of the two constituent monomers, which can be calculated using the Reuss model.(ii)The failure mode of the composite specimen is mainly controlled by the height–diameter ratio of the weak body. The weak body breaks, and the strong body rebounds elastically as a kind of loading acting on the weak body, which may increase the post-peak strain rate of the weak body. When the height–diameter ratio of the weak body is small, it mainly shows splitting failure, and when the height–diameter ratio is large, it shows shear failure. When the height–diameter ratio is 1~2, it is a mixed mode of splitting and shear fracturing.(iii)For impact propensity, it can be determined that the uniaxial compressive strength of the combination is higher than that of the weak body, and the dynamic failure time is lower than that of the weak body. It is impossible to determine the variations of the elastic energy index and impact energy index of the combination with respect to that of the weak body.

Coal and rock contain pores and cracks, which lead to significant heterogeneity and anisotropy. Data obtained in the laboratory test using raw rock samples are often discrete, which has a greater impact on the analysis and results of coal–rock composite research, which should be considered.

## Figures and Tables

**Figure 1 materials-16-03681-f001:**
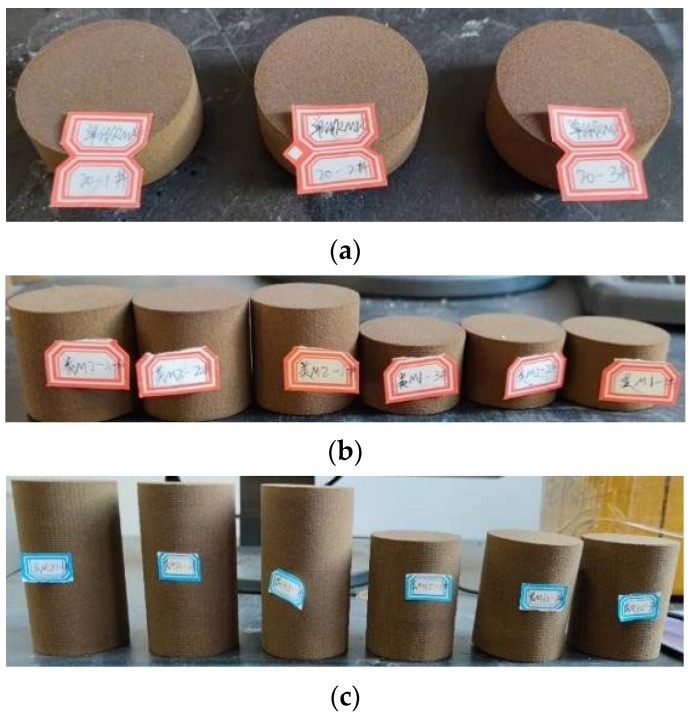
Three-dimensional-printed coated sand specimens, (**a**) the heights are 20 mm, (**b**) the heights are 50 mm and 33 mm, (**c**) the heights are 80 mm and 66 mm.

**Figure 2 materials-16-03681-f002:**
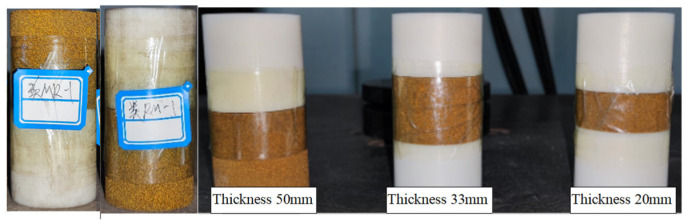
Combined specimens after 3D printing.

**Figure 3 materials-16-03681-f003:**
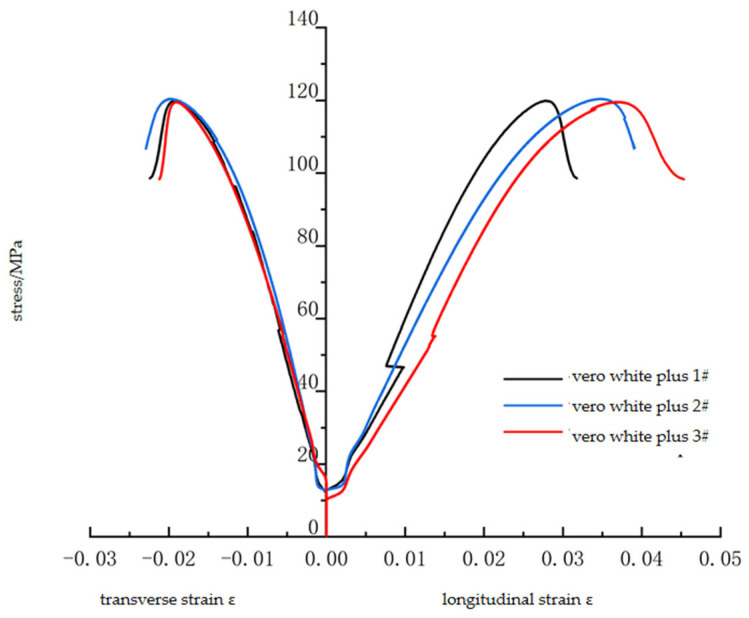
Uniaxial compression test results of standard Vero White Plus specimens.

**Figure 4 materials-16-03681-f004:**
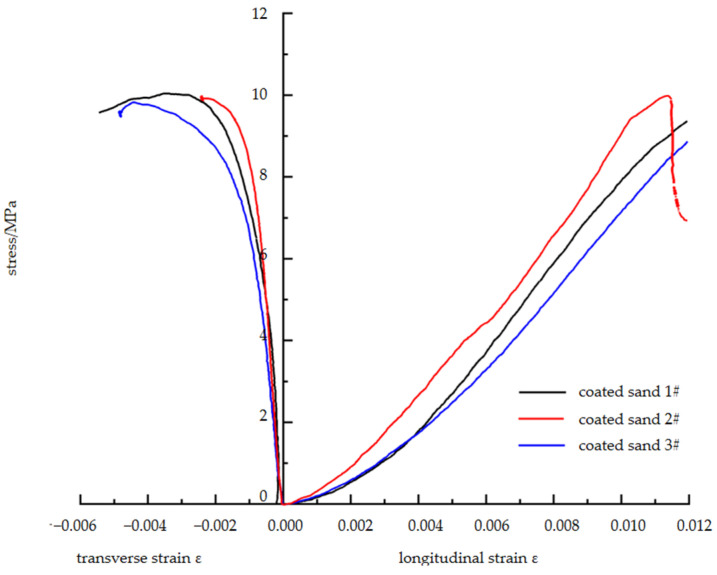
Uniaxial compression test results of standard coated sand specimens.

**Figure 5 materials-16-03681-f005:**
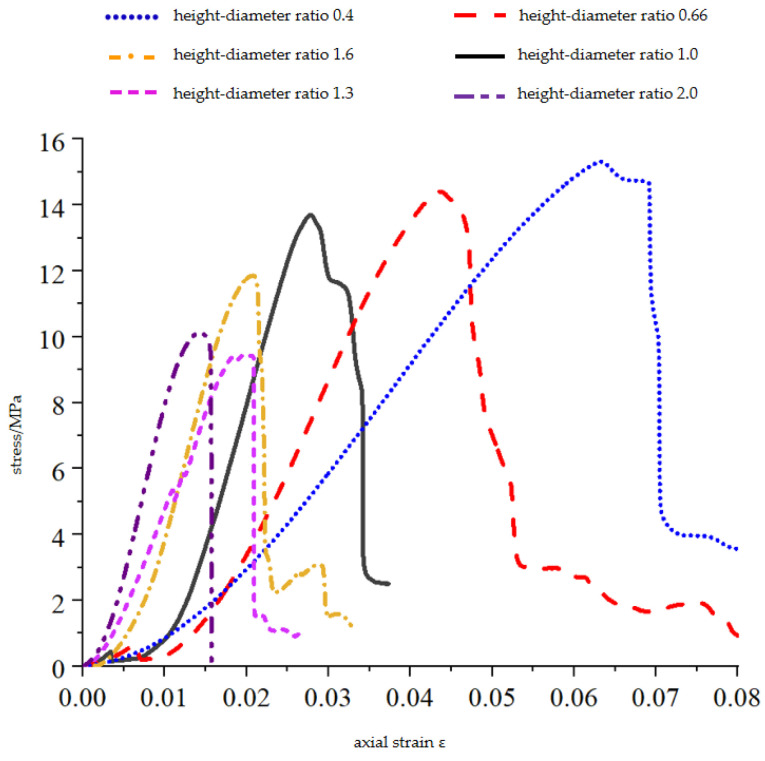
Complete compressive stress–strain curves of coated sand specimens with different length-to-diameter ratios.

**Figure 6 materials-16-03681-f006:**
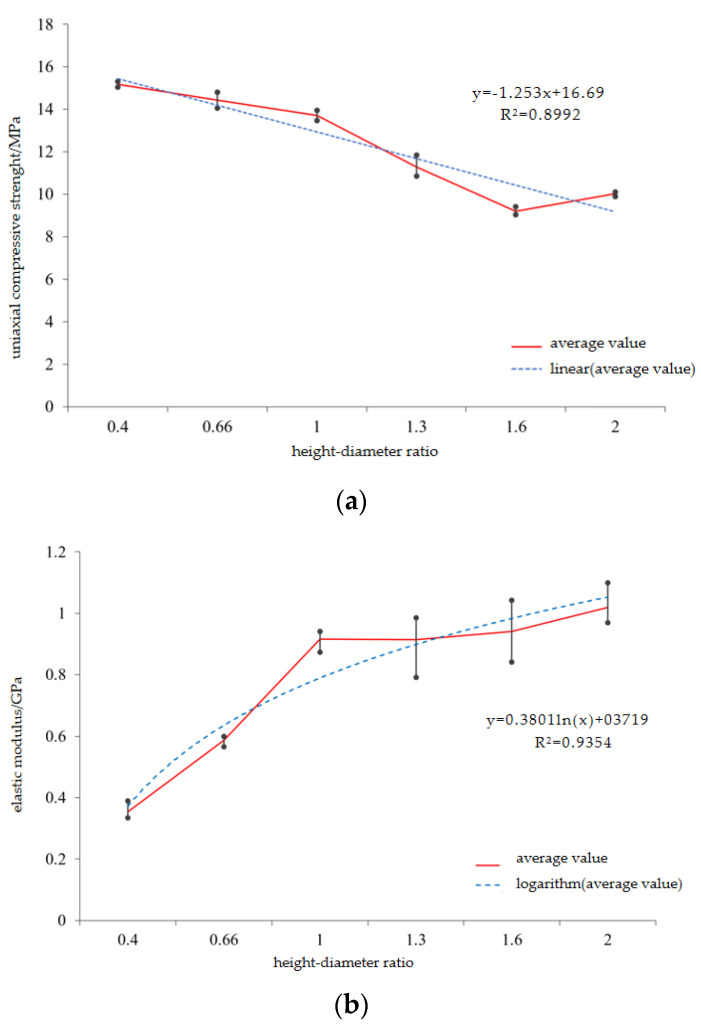
Relationship between *UCS*, *E*, and *ε1* and length-to-diameter ratio of specimens, (**a**) relationship between uniaxial compressive strength and height–diameter ratio, (**b**) relationship between elastic modulus and height–diameter ratio.

**Figure 7 materials-16-03681-f007:**
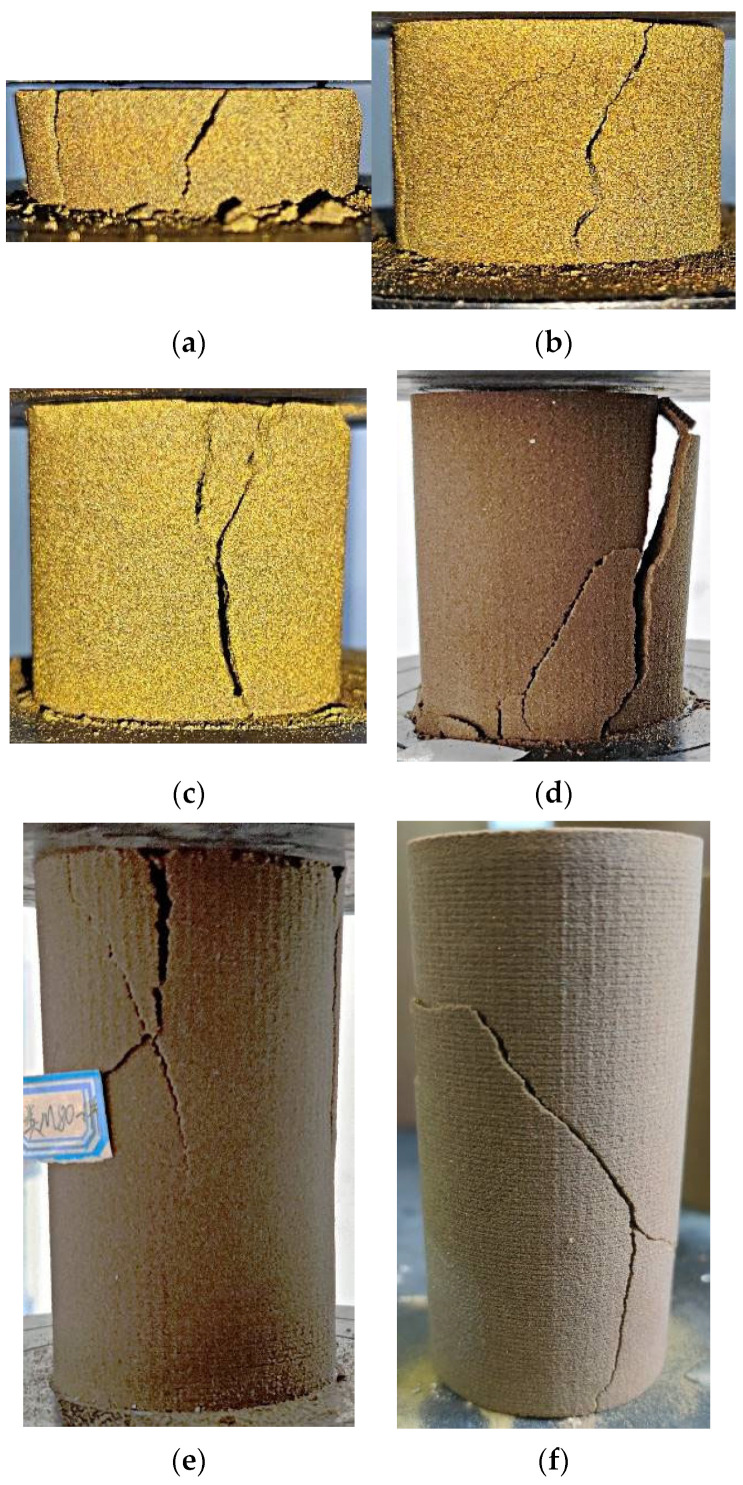
Fracture characteristics of coated sand specimens with different length-to-diameter ratios, (**a**) height–diameter ratio of 0.4, (**b**) height–diameter ratio of 0.66, (**c**) height–diameter ratio of 1, (**d**) height–diameter ratio of 1.3, (**e**) height–diameter ratio of 1.6, (**f**) height–diameter ratio of 2.

**Figure 8 materials-16-03681-f008:**
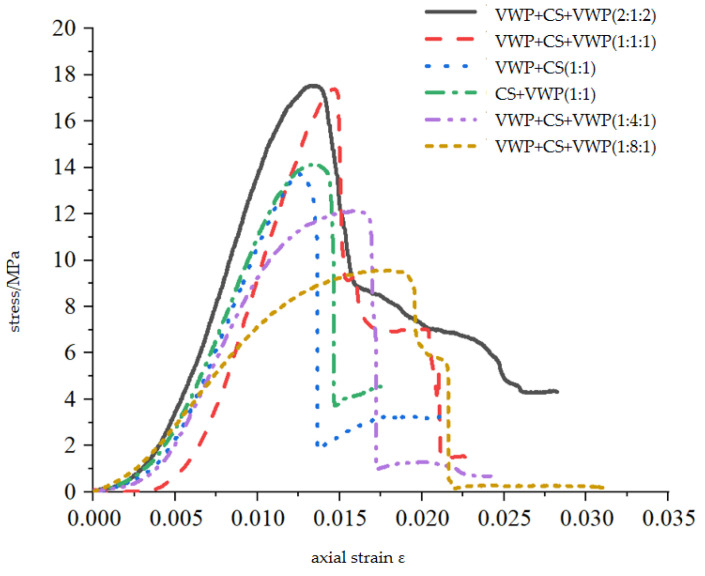
Complete compressive stress–strain curves of combined specimens.

**Figure 9 materials-16-03681-f009:**
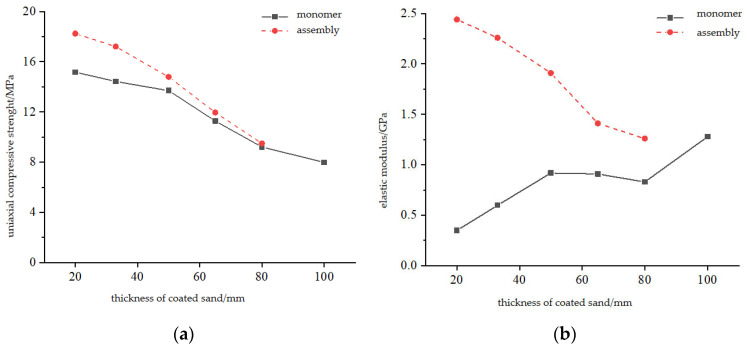
Relation between UCS, E, and thickness of coated sand, (**a**) uniaxial compressive strength of coated sand monomer and coated sand monomer, (**b**) elastic modulus of group and coated sand monomer.

**Figure 10 materials-16-03681-f010:**
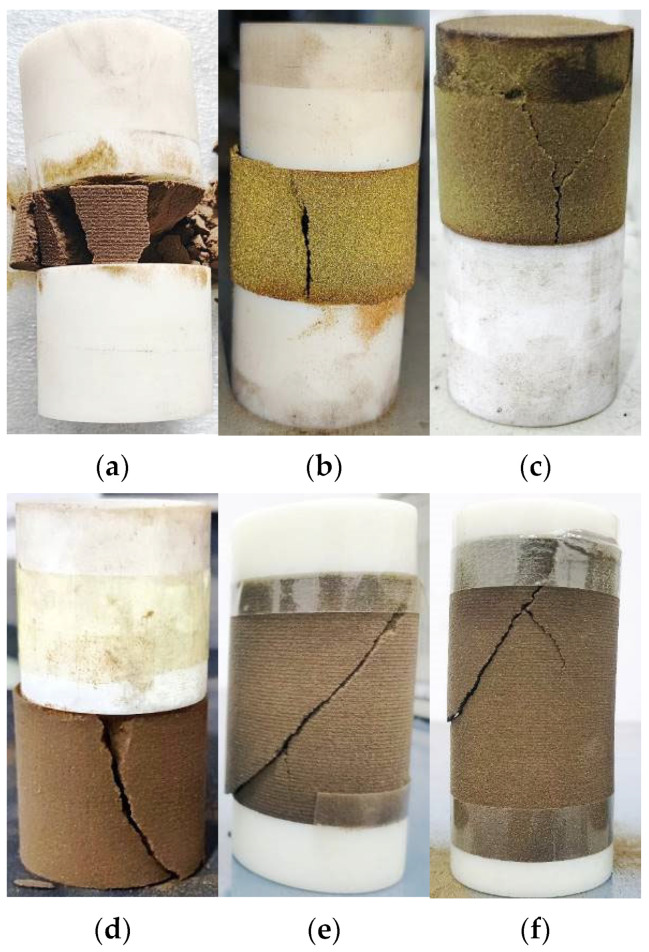
Failure characteristics of combined specimens, (**a**) VWP + CS + VWP, (2:1:2), (**b**) VWP + CS + VWP, (1:1:1), (**c**) VWP + CS, (1:1), (**d**) CS + VWP, (1:1), (**e**) VWP + CS + VWP, (1:4:1), (**f**) VWP + CS + VWP, (1:8:1).

**Figure 11 materials-16-03681-f011:**
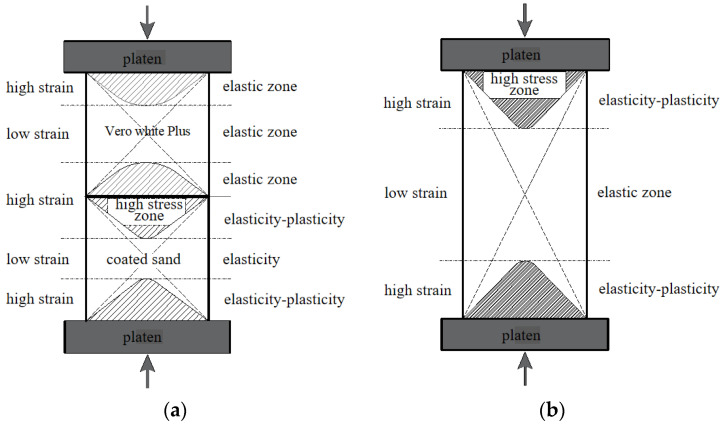
Conceptual explanation for the stress and strain of combined and single specimen, (**a**) combined specimen with height–diameter ratio of 2, (**b**) monomer sample with height–diameter ratio of 2.

**Figure 12 materials-16-03681-f012:**
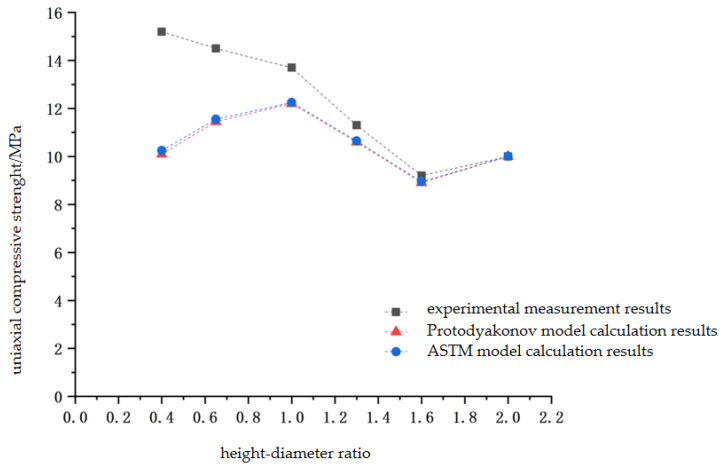
Empirical Formula calculation and laboratory test of combined specimen’s UCS.

**Figure 13 materials-16-03681-f013:**
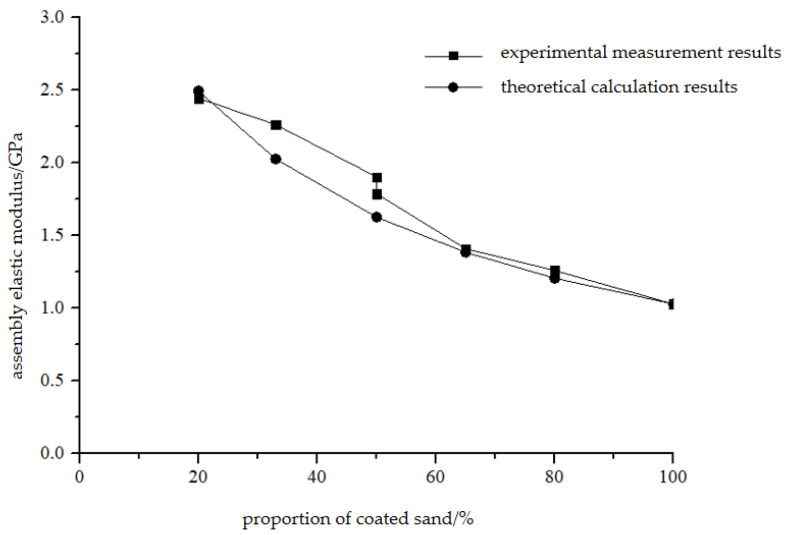
Theory calculation and laboratory test of combined specimen’s E.

**Figure 14 materials-16-03681-f014:**
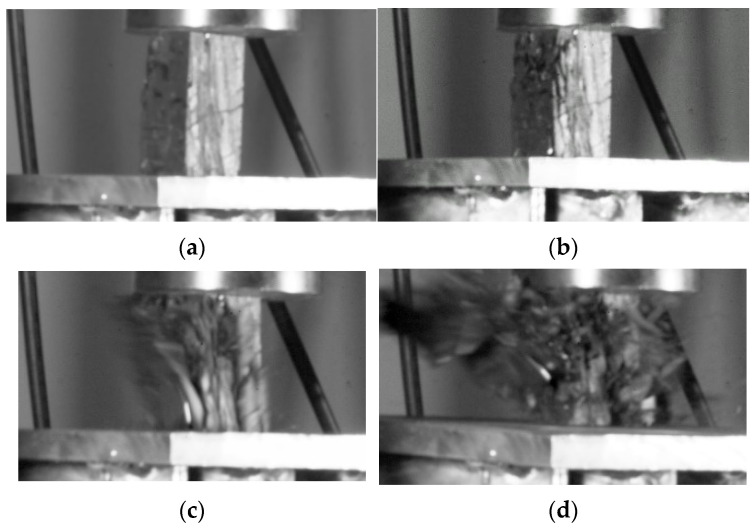
Coal specimen bursting under soften loading, (**a**) 1.02 s, (**b**) 2.85 s, (**c**) 3.34 s, (**d**) 3.37 s.

**Figure 15 materials-16-03681-f015:**
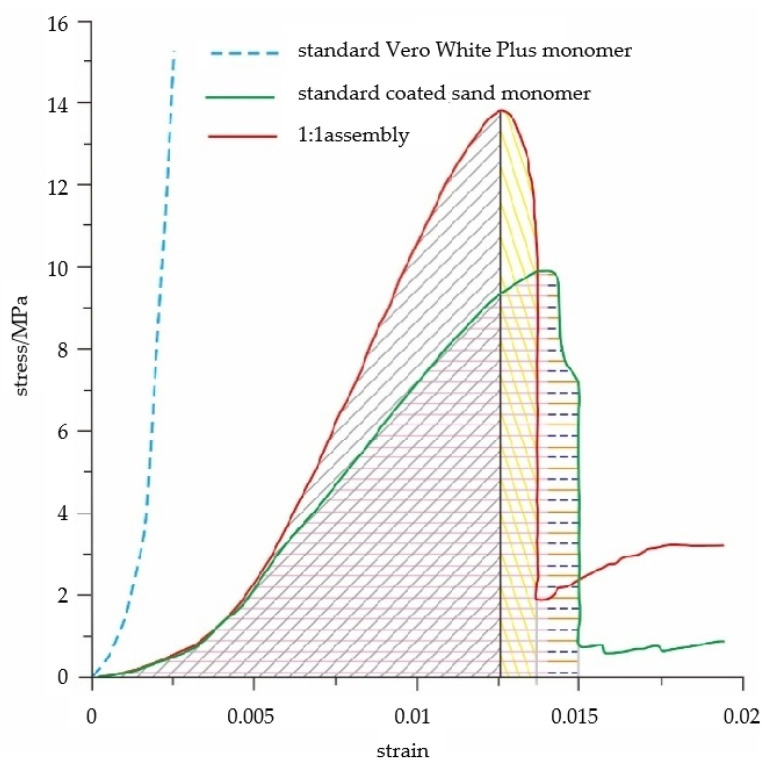
Energy comparison between combination and units under uniaxial compression. The shaded parts is for calculation of the energy before and after the peak strength of the specimen.

**Table 1 materials-16-03681-t001:** Results of standard 3D-printed specimens.

Material	No.	*UCS* [MPa]	*E* [GPa]	ν
Vero White Plus	VWP-1#	119.85	3.821	0.49
	VWP-2#	120.33	4.443	0.48
	VWP-3#	119.50	3.349	0.37
	Ave.	119.89	3.871	0.44
Coated Sand	CS-1#	10.12	1.114	0.09
	CS-2#	9.89	1.002	0.10
	CS-3#	10.04	0.971	0.11
	Ave.	10.02	1.029	0.10

**Table 2 materials-16-03681-t002:** Uniaxial compression test of coated sand specimens with different height-to-diameter ratios.

Height: Diameter	No.	*UCS* [MPa]	*E* [GPa]
0.4	CS04-1#	15.16	0.336
	CS04-2#	15.06	0.389
	CS04-3#	15.31	0.334
	Ave.	15.18	0.353
0.66	CS04-1#	14.41	0.565
	CS04-2#	14.82	0.599
	CS04-3#	14.05	0.598
	Ave.	14.43	0.587
1	CS10-1#	13.48	0.941
	CS10-2#	13.95	0.934
	CS10-3#	13.70	0.873
	Ave.	13.71	0.916
1.3	CS13-1#	11.10	0.792
	CS13-2#	11.86	0.986
	CS13-3#	10.87	0.967
	Ave.	11.28	0.915
1.6	CS16-1#	9.16	1.042
	CS16-2#	9.05	0.842
	CS16-3#	9.43	0.937
	Ave.	9.21	0.940
2	CS20-1#	10.12	1.114
	CS20-2#	9.89	1.002
	CS20-3#	10.04	0.971
	Ave.	10.02	1.029

**Table 3 materials-16-03681-t003:** Uniaxial compression test results of 3D-printed composite specimens.

Composite Method	No.	*UCS* [MPa]	*E* [GPa]
VWP + CS + VWP (2:1:2)	C20-1#	18.43	2.441
	C20-2#	18.79	2.645
	C20-3#	17.53	2.249
	Ave.	18.25	2.443
VWP + CS + VWP (1:1:1)	C33-1#	17.39	2.327
	C33-2#	17.72	2.394
	C33-3#	16.53	2.061
	Ave.	17.21	2.262
VWP + CS (1:1)	C50-1#	15.03	1.863
	C50-2#	15.39	1.812
	C50-3#	14.62	2.031
	Ave.	15.01	1.902
CS + VWP (1:1)	C50-1#	14.15	1.755
	C50-2#	14.65	1.754
	C50-3#	14.85	1.853
	Ave.	14.55	1.787
VWP + CS + VWP (1:4:1)	C65-1#	11.94	1.332
	C65-2#	12.12	1.476
	C65-3#	11.81	1.421
	Ave.	11.96	1.409
VWP + CS + VWP (1:8:1)	C80-1#	9.35	1.233
	C80-2#	9.53	1.402
	C80-3#	9.61	1.144
	Ave.	9.50	1.259

**Table 4 materials-16-03681-t004:** Elastic modulus comparison between calculated and measured results of combined and unit specimens.

Ref.	*EM* _C_	*EM* _R_	*f_C_*	*f* _R_	*EM* Measured	*EM* Calculated
[17]	2.51	5.66	50	50	3.39	3.48
[17]	2.51	6.48	50	50	3.54	3.62
[17]	2.51	12.54	50	50	4.04	4.18
[17]	2.51	21.65	50	50	4.35	4.50
[17]	2.51	30.98	50	50	4.48	4.64
[19]	2.51	5.05	25	75	4.52	4.03
[19]	2.51	5.05	33	67	4.03	3.79
[19]	2.51	5.05	50	50	3.19	3.35
[19]	2.51	5.05	67	33	2.99	3.01

Note: *EM*_C_—coal elastic modulus; *EM*_R_—elastic modulus of rock mass; *f_C_*—volume integral of coal; *f*_R_—volume integral of rock.

**Table 5 materials-16-03681-t005:** Burst proneness of combined specimen and single specimen.

	Indicator	Monomer	Composite	Ref.
1	Uniaxial compressive strength	Low	High	Figure 9a
2	Elastic energy index	-	-	
3	Impact energy index	-	-	
4	Dynamic destruction time	Large	Small	Figure 15

## Data Availability

The data used to support the findings of this study are available from the corresponding author upon reasonable request (chenying@lntu.edu.cn).

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
