# Peer review of "Experimental Study of Mechanical Properties and Failure Characteristics of Coal–Rock-like Composite Based on 3D Printing Technology"

_materials, 2023, doi:10.3390/ma16103681_

Round 1

Reviewer 1 Report

The article contains interesting laboratory research that relates to physical modeling of coal and surrounding rocks. The performed sample models as well as strength and deformation tests are interesting due to the obtained stress-strain characteristics, in particular for various configurations. I recommend this article for acceptance for publication in Materials after implementing the necessary corrections.

1. In the introduction, some information about commonly used materials to model coal and surrounding rocks should be added;

2. For the second Figure, it should be added a simple diagram (2c), on which the dimensions will be marked;

3. In the subsection 2.2, it should be written what type of strain gauges and amplifier was used in the tests;

4. For the results in Table 1, it should be written how many samples were tested for each batch;

5. In the third chapter, it should be added information regarding the bulk density of the samples and try to find the relationship between compressive strength and density, especially on samples composed of different materials;

6. In the subsection 4.1.1 for Figure 11, it should be clearly stated whether the presented stress distribution diagrams result from the conducted tests or rather it is a reference to the literature;

7. Figure 12, markings for Protodyakonov and ASTM model are completely invisible, they should be marked with a different color to distinguish them from experimental results;

8. In the subsection 4.2, table 5, a descriptive classification was proposed: low, high, large, small - therefore, these values should be described in the text, in particular what are the ranges for which the appropriate value is included;

9. One statement regarding dynamic loads should be added to the conclusions.

Author Response

Attachment

Reviewer 2 Report

Line 42: many references [1-16]. Please elaborate.

Figure 1. captions “height is” -> “heights are”

“3D printing combined specimens” -> “Combined specimens after 3D printing”

Figure 2. please mark Vero White 1,…, in the figure. Otherwise, figure 3 does not make sense.

Please draw a schematic showing the load direction, traverse, longitudinal, and axial strain. 

Figure 6. the “x” and “y” in the equations do not make sense. Use the parameters of the abscissa and ordinates instead.

Equations 1-3. it is weird. Please use the equation editor and write properly.

Instead of a stress versus strain curve, having at least one load versus deformation curve is better. Then, a stress versus strain curve can follow it. 

Units are shown in “[]” not by “/”, e.g., stress [MPa]

Author Response

Attachment

Reviewer 3 Report

Please refer to the attached file

Author Response

Attachment

Round 2

Reviewer 1 Report

The article has been well improved.

Author Response

Thank you.

Reviewer 3 Report

The manuscript has been improved but still has a few comments that need to amend accordingly before being accepted for publishing.
Please refer to the attached file.
